# Deep learning predictions of TCR-epitope interactions reveal epitope-specific chains in dual alpha T cells

Giancarlo Croce [1,2,3,4], Sara Bobisse[3,4,5], Dana Léa Moreno[1,2,3,4], Julien Schmidt[1,4,5], Philippe Guillame[4,5], Alexandre Harari [1,3,4,5] & David Gfeller [1,2,3,4] ✉

T cells have the ability to eliminate infected and cancer cells and play an essential role in cancer immunotherapy. T cell activation is elicited by the binding of the T cell receptor (TCR) to epitopes displayed on MHC molecules, and the TCR specificity is determined by the sequence of its α and β chains. Here, we collect and curate a dataset of 17,715 αβTCRs interacting with dozens of class I and class II epitopes. We use this curated data to develop MixTCRpred, an epitope-specific TCR-epitope interaction predictor. MixTCRpred accurately predicts TCRs recognizing several viral and cancer epitopes. MixTCRpred further provides a useful quality control tool for multiplexed single-cell TCR sequencing assays of epitope-specific T cells and pinpoints a substantial fraction of putative contaminants in public databases. Analysis of epitope-specific dual α T cells demonstrates that MixTCRpred can identify α chains mediating epitope recognition. Applying MixTCRpred to TCR repertoires from COVID-19 patients reveals enrichment of clonotypes predicted to bind an immunodominant SARS-CoV-2 epitope. Overall, MixTCRpred provides a robust tool to predict TCRs interacting with specific epitopes and interpret TCR-sequencing data from both bulk and epitope-specific T cells.

T cells are key components of the cellular immune response, providing defense against infected and malignant cells. In cancer, inducing new T cell responses or boosting pre-existing ones has revolutionized cancer immunotherapy treatments, providing long-term benefits to a significant fraction of patients, including some with late-stage malignancies[1–3]. The activation of a T cell is triggered by the binding of the T cell receptor (TCR) to antigen-derived peptides that are presented on major histocompatibility complex molecules (pMHCs). TCRs have an extensive sequence diversity, with estimates ranging from $10^{15}$ to $10^{61}$ different TCR sequences that can potentially be generated[4–6]. This high diversity allows T cells to recognize a large number of epitopes displayed on different MHC alleles[7]. As of today, high-throughput sequencing enables researchers to rapidly map TCR repertoires in patients[8,9]. However, it remains challenging to know which TCRs target specific epitopes. This hinders the development of treatments that aim at using or engineering T cells to target specific peptides displayed on MHC molecules, such as cancer neoepitopes[10,11].

TCRs are heterodimers composed of one α and one β chain. The TCR sequence diversity is achieved during the V(D)J recombination

[1]Department of Oncology UNIL CHUV, Ludwig Institute for Cancer Research, University of Lausanne, Lausanne, Switzerland. [2]Swiss Institute of Bioinformatics (SIB), Lausanne, Switzerland. [3]Agora Cancer Research Centre, Lausanne, Switzerland. [4]Swiss Cancer Center Leman (SCCL), Lausanne, Switzerland. [5]Department of Oncology UNIL CHUV, Ludwig Institute for Cancer Research, University Hospital of Lausanne, Lausanne, Switzerland. ✉e-mail: david.gfeller@unil.ch

when a unique combination of V and J (for the α chain) and V, D, and J (for the β chain) germline-encoded segments is selected and assembled. Additional diversity occurs through N- and P- nucleotide insertions at the V(D)J junctions. These regions, referred to as complementarity-determining regions 3 (CDR3), are mainly involved in the recognition of the epitope, while two other CDRs (CDR1 and CDR2) located on the V segments mediate contact primarily with the MHC[12].

Most T cells express a unique α and a unique β chain[13]. However, T cells expressing two in-frame-rearranged TCRα or two TCRβ chains have been observed both in *Mus musculus* and *Homo sapiens*[14–20]. It is currently estimated that approximately 10% of T cells can express two functionally rearranged α-chains, whereas dual β chains are found in less than 1% of T cells[19,21–25]. Many TCR-sequencing analysis tools disregard dual chain T cells[26] or consider the most expressed chain as the one responsible for epitope recognition[27].

Several immune assays have been developed to isolate epitope-specific T cells and sequence the α and β chains of their TCRs[7,28,29]. Many approaches use individual pMHC multimers to sort and sequence T cells recognizing one specific epitope[30,31]. Recently, the throughput of such approaches has been expanded by taking advantage of multiplexed DNA barcoded pMHC multimers coupled with single-cell TCR/barcode-sequencing[32]. Another approach to enhance the number of epitopes that can be simultaneously analyzed consists of stimulating pools of T cells with various combinations of epitopes and deconvolving the different pools[33]. Conventional bulk sequencing methods have been applied to sequence the TCRs in epitope-specific T cell populations one chain at a time or only the β chain. More recently, single-cell TCR sequencing has enabled the acquisition of paired αβTCR sequences. This is particularly relevant for modeling TCRs of epitope-specific T cells since both chains ultimately determine the TCR specificity[27,34–37]. One of the most comprehensive αβTCR sequence datasets of epitope-specific T cells[29,38] was generated by the 10x Genomics immune profiling platform, coupling DNA barcoded pMHCs multimers with single-cell TCR-Seq[38]. This study identified approximately 15,000 TCR clonotypes interacting with 44 epitopes in one single experiment[38]. These data constitute around 70% of all paired αβTCR–epitopes currently stored in public databases[29,39]. Although recently developed quality control tools suggest that not all these interactions are of equal quality[27,40,41], this type of technology is likely to play an important role in providing information about the specificity of TCRs recognizing distinct viral or cancer epitopes.

Paired TCR–pMHCs sequence data have been used to train machine learning approaches that aim at predicting which T cells can target a specific pMHC directly from the TCR sequences[42,43]. While some tools consider only the CDR3 region and/or the V, J segments of the β chain (e.g., TITAN[44], ATM-TCR[45], ImRex[46], pMTnet[47], and TCRex[48,49]), others take as input the full sequence of the TCR (i.e., the α and the β chain)[27,35,36,50–55]. These methods range from distance-based classifiers[50,52,56] to machine learning or deep learning models[27,35–37,49,53,54,57,58], and they all share the common underlying assumption that TCRs displaying similar sequence patterns recognize the same pMHC[50,59]. A fraction of these tools can be used directly through command-line or web interfaces (e.g., NetTCR2.1[35], ERGO2.0[37,54], tcrdist3[50]), while others need to be retrained (e.g., TCRAI[27] and TCRGP[57]), or have been benchmarked[43] but not yet released (e.g., TCRex[48,49] for αβ TCRs, SONIA[53] for TCR classification). Most TCR–epitope interaction predictors have been trained and tested for predicting TCRs recognizing specific pMHCs with at least some known TCRs (referred to as epitope-specific predictions), although some tools include in theory, the possibility to make predictions for TCRs recognizing any epitope (referred to as pan-epitope predictions). Due to different training data and procedures, it is challenging to compare the advantages and disadvantages of each approach. To

address this issue, a public benchmark for TCR–pMHC predictions was recently introduced[43].

Several conclusions can be drawn from these studies. First, using paired α and β chains is important for modeling TCR specificity[27,34–37,43], and predictors that rely on one of the chains (usually the β chain) have been shown to be less accurate than methods using both chains[43]. Second, algorithms employing different approaches, from distance-based to deep learning methods, have comparable performance[43]. Third, accurate predictions require a minimum number of TCRs interacting with a specific pMHC. This demonstrates that a key determinant of TCR–pMHC interaction predictions is the quality and epitope coverage of the training set. It further suggests that extrapolating these predictions to pMHC without known interacting TCRs is challenging[35,43].

In this study, we collect and curate a large dataset of paired αβTCR sequences coupled with their cognate pMHC. We leverage these data to develop a sequence-based predictor of TCR–pMHC interaction, referred to as MixTCRpred (Fig. 1A). We show that MixTCRpred can accurately predict TCRs binding to several known viral and cancer epitopes, outlines how much predictions can be extended to new epitopes, serves as a valuable control tool for identifying putative contaminants in existing databases, allows accurate annotation of epitope-specific chains in dual α T cells, and reveals enrichment of TCRs predicted to recognize an immunodominant class II epitope in TCR repertoires of COVID-19 patients (Fig. 1A).

## Results

### Integration and curation of αβTCR–pMHCs interactions reveal binding specificities for dozens of class I and class II epitopes

To improve our understanding of the specificity of TCRs for different epitopes, we collected sequences of αβTCRs targeting specific pMHCs from several public databases, including VDJdb[29], IEDB[39], and the McPAS database[60] (Fig. 1B and "Methods"). TCR–pMHC sequence data from the 10x Genomics immune profiling assay[38] were processed separately to include only cases with a clear signal from one unique pMHC multimer (see "Methods"). We further collected and curated TCRs isolated from *Mus musculus* infected with *Lymphocytic choriomeningitis* (LCMV) from two recent studies[61,62]. Duplicated TCR–pMHC pairs were removed based on V/J gene usage as well as the same CDR3 sequence for both the α- and β-chains. This led to a total of 20,279 distinct TCR sequences interacting with 1253 pMHCs (Fig. 1B). For the majority of pMHCs, only one or a few TCRs have been experimentally validated, leading to a heavily skewed TCR distribution (Fig. 1C, D). For further analysis, only pMHCs with at least ten binding TCRs were considered, resulting in a total of 17,715 αβTCRs interacting with 146 pMHCs (Fig. 1C and Source Data file). As expected, most of the data relates to peptides presented by human MHCs (127 out of 146 pMHCs), and with a large fraction of HLA-A*02:01 restricted peptides (Fig. 1E). In contrast, only 19 pMHCs in *Mus musculus* have been extensively characterized in terms of TCRs. One important case is the class II LCVM-derived peptide, DIYKGVYQFKSV restricted to H2-IAb, for which 3650 different αβTCRs obtained from 11 LCMV infected mice were available from two different studies[61,62].

### MixTCRpred accurately predicts TCRs recognizing specific pMHCs

We used the collected data to train and validate MixTCRpred, a machine learning predictor of TCR–pMHC interactions (Fig. 2A). MixTCRpred is a pMHC-specific predictor, where a separate model is trained for each epitope. Negative data were computationally generated by sampling TCRs with different or undetermined specificity (see "Methods"). For each pMHC we chose a ratio 1:5 between positives (epitope-specific TCRs) and negatives (non-binding TCRs). A list of the available MixTCRpred models is provided in the Source Data file.

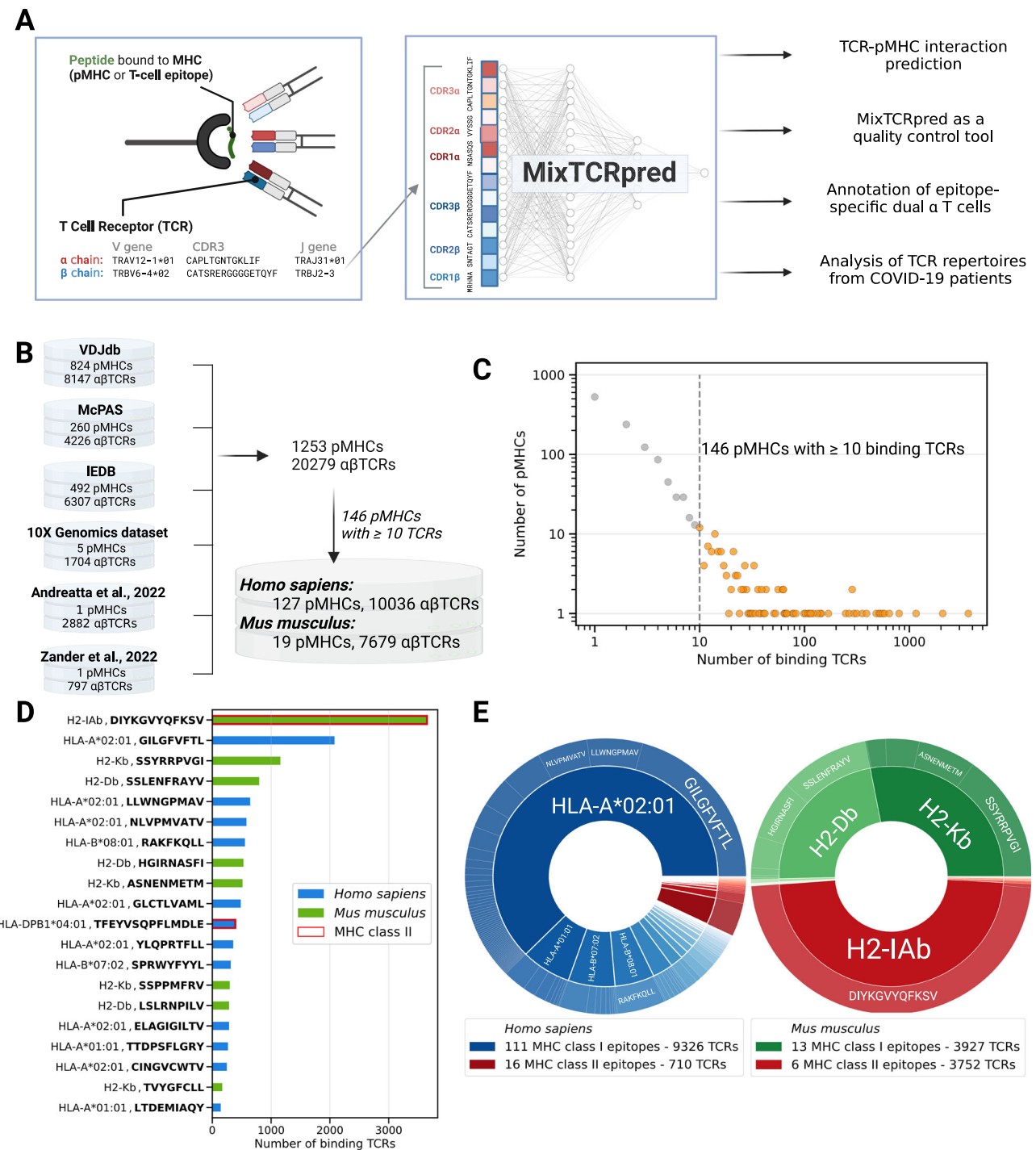

**Fig. 1 | Integration and curation of αβTCR−pMHCs interactions reveal binding specificities for dozens of class I and class II epitopes. A** Overview of our pipeline, including data collection, training of MixTCRpred, and applications. **B** Summary of the datasets collected in this study with the corresponding number of TCRs and pMHCs. **C** Distribution of pMHCs interacting with different numbers of TCRs. One-hundred forty-six pMHCs have 10 or more experimentally validated binding αβTCRs, with a total of 17,715 αβTCRs. **D** Barplots showing the number of αβTCRs for the top 20 pMHCs with the most experimentally validated αβTCRs. **E** Distribution of TCRs recognizing epitopes restricted to different MHC alleles. Source data are provided as a Source Data file.

The architecture of MixTCRpred is depicted in Fig. 2A. From an input TCR, which is conventionally provided as V, J genes, and CDR3 sequences for both chains, MixTCRpred first extracts the CDR1, CDR2 sequences (from the V genes) as defined by the International ImMunoGeneTics Information System[63]. The CDR1, CDR2, and CDR3 sequences are then padded and concatenated for the α and β chains separately. The TCR sequences are numerically embedded through a machine learning-based embedding, where embedding vectors are initially sampled randomly and then adjusted in the course of the training. A transformer encoder[64] is then used to identify the statistical patterns underlying the TCR specificity (see "Methods"). The final step is a dense classification layer whose output score indicates how likely the TCR is to interact with a specific pMHC. Comparing raw scores across different epitopes is challenging due to the inherent biases of each model. To make the predictions more interpretable and comparable, we developed a robust framework to compute the %rank,

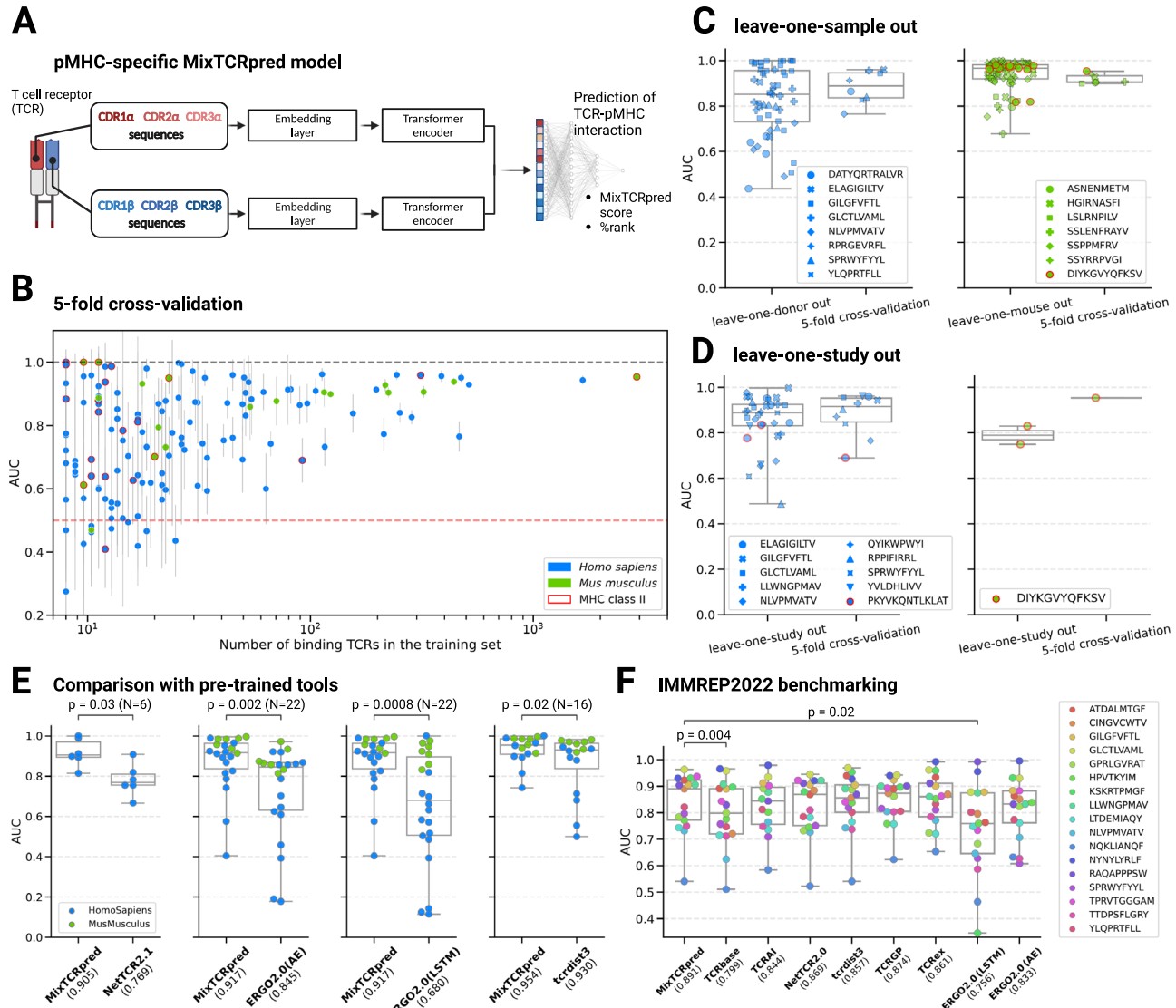

**Fig. 2 | MixTCRpred accurately predicts TCRs recognizing specific pMHCs.**
**A** Illustration of the MixTCRpred model architecture. For each pMHC, MixTCRpred predicts if a TCR (encoded based on the CDR1, CDR2, and CDR3 α- and β sequences) would target it. The outputs are the predicted MixTCRpred interaction score and the corresponding % rank. **B** 5-fold cross-validation average AUCs for the 146 pMHCs included in our dataset. The vertical lines show the standard deviations. The dashed lines correspond to random (red line, AUC of 0.5) and perfect predictions (black line, AUC of 1). **C** Comparison of the AUC values in leave-one-sample-out cross-validation (one point for each sample and each pMHC) with the average AUC values in the 5-fold cross-validations (one point for each pMHC). N = 73 independent samples were used in the leave-one-sample-out cross-validation for *Homo Sapiens* and N = 56 for *Mus Musculus*. **D** Comparison of the AUC values in leave-one-study-out cross-validation (one point for each study and each pMHC)

with the average AUC values in the 5-fold cross-validations (one point for each pMHC). N = 33 independent studies were used in the leave-one-study-out cross-validation for *Homo Sapiens* and N = 2 for *Mus Musculus*. **E** Comparison of MixTCRpred with other pre-trained tools for *Homo sapiens* and *Mus musculus* pMHCs. The median AUC for each tool is included in parentheses. For each tool separately, the statistical comparison with MixTCRpred was performed considering the pMHCs supported by both methods. **F** AUCs of MixTCRpred and other tools for the 17 pMHCs of the IMMREP22 benchmark dataset. The center line within the box represents the median value, with the bottom and top bounds of the box delineating the 25th and 75th percentiles, respectively. Whiskers extend to minimum and maximum values. The p-values were obtained with a two-sided paired t-test. Source data are provided as a Source Data file.

which indicates how the raw score of a given TCR compares with that of a large set of randomly generated TCRs (see "Methods" and Supplementary Fig. 1).

To explore the accuracy of our predictions and the impact of the size of the training set, we first performed a standard 5-fold cross-validation for each of the 146 MixTCRpred models. Performance was assessed with the Area Under the receiver operating curve (AUC) (see the Source Data file). As shown in Fig. 2B, MixTCRpred models achieved robust predictions for pMHCs with a large number of interacting TCRs: out of 43 pMHCs with more than 50 TCRs, 40 had an

average AUC > 0.7, and 34 of them an AUC > 0.8. Lower accuracy was observed for several pMHCs with fewer TCRs.

We next performed a leave-one-sample-out cross-validation to determine whether MixTCRpred predictions were consistent across samples within the same study. 15 epitopes had been analyzed in multiple samples, each of which had at least 10 TCRs. Figure 2C shows that in the majority of the cases, MixTCRpred was successful in predicting epitope-specific TCRs in a new sample, with leave-one-sample-out performances similar to that of a 5-fold cross-validation. To assess whether predictions could be transferred across different

studies, we performed a leave-one-study-out validation, including epitopes with at least 10 TCRs per study. Overall, we observed only limited loss in predictive power with respect to the AUCs of 5-fold cross-validations on the same datasets (Fig. 2D). This demonstrates that MixTCRpred predictions are robust across studies and could be applied to new studies for the epitopes considered in the MixTCRpred training set.

This conserved predictive power could result from either conservation of TCR sequence patterns captured by MixTCRpred, or the presence of public clones that are found across different samples/ studies. To shed light on this question, we focused on TCRs specific for the class II H2-IAb, DIYKGVYQFKSV epitope from LCMV-infected *Mus Musculus* samples for which multiple samples from 2 studies were available (Supplementary Fig. 2). Out of a total of 11 samples, we observed that most TCRs are unique to one sample (approximately 98% of the epitope-specific TCR repertoire), and only a limited number of TCR sequences are shared across two or more samples (Supplementary Fig. 2B, C). Despite this low clonal overlap, the CDR3α and CDR3β display similar motifs across different samples and studies (Supplementary Fig. 2D). These epitope-specific TCRs came from mono-allelic *Mus Musculus* strains elevated in controlled conditions. Therefore, most of the observed TCRs variability is attributable to the fact that a large number of different TCRs can be generated to target the same pMHC as long as they satisfy the statistical constraints reflected by the sequence patterns and captured by MixTCRpred. Similar results were obtained for other epitopes (Supplementary Fig. 3).

## MixTCRpred compares favorably with other TCR–pMHC interaction prediction tools

Next, we benchmarked MixTCRpred with other publicly available tools that take αβTCRs as input. First, we evaluated the performance of our predictor against the other available pre-trained models accessible through command-line or web interfaces, i.e., NetTCR2.1[35], ERGO2.0 (AE), ERGO2.0 (LSTM)[37,54] and tcrdist3[50] (see "Methods"). To this end, we used the McPAS database as a test set, which is not part of the training dataset of most tools considered in this validation with the exception of NetTCR2.1. MixTCRpred was retrained excluding data from this database as well as overlap in other databases (see "Methods"). A comparison of the performance was done for the set of pMHCs that were supported by each tool in their pre-trained version. Our results demonstrate that MixTCRpred consistently outperforms other available tools for *Homo sapiens* and *Mus musculus* pMHCs (Fig. 2E).

To extend our benchmark to other methods, including some that have not yet been released, we capitalized on the recent IMMREP22 dataset[43] consisting of curated data for 17 peptides–MHC, each having at least 50 unique validated binding αβTCR sequences. This dataset was specifically collected to benchmark the algorithms behind TCR–pMHC interaction predictors (i.e., using the same training and test sets for all methods). Upon retraining our tool on the same training set as all other tools, we observed that MixTCRpred achieved similar or higher accuracy on the test set (median AUC of 0.891, Fig. 2F). This indicates that the architecture of MixTCRpred provides state-of-the-art performance, even when not considering our efforts to enhance and curate the training set.

We further used the IMMREP22 dataset to assess the role of the CDR1 and CDR2 sequences in MixTCRpred (Supplementary Fig. 4A). We observed that predictions with the CDR1, CDR2, and CDR3 as input features are more accurate than those obtained using only the CDR3 for most pMHCs (Supplementary Fig. 4B), which is consistent with previous observations[35,43]. Overall, our results show that MixTCRpred achieves robust predictions for pMHCs for which several interacting TCRs have been experimentally determined.

## MixTCRpred reveals how much predictions can be extended to unseen epitopes

To investigate whether predictions may be extended to epitopes not present in the training set, we adapted MixTCRpred architecture to incorporate both the peptide and TCR sequences as inputs, resulting in a so-called pan-epitope predictor. This involved adding an extra embedding and transformer encoder layer for the epitope sequence and concatenating it with the TCR before the final classification layer (Supplementary Fig. 5). By doing so, the model is, in theory, able to learn correlation patterns between the TCR and epitope sequences, and potentially predict TCRs binding to epitopes without any known TCR (i.e., unseen epitopes)[35]. To avoid overly complex models, we trained a separate pan-epitope MixTCRpred model for each MHC allele.

To evaluate the performance of the pan-epitope version of MixTCRpred, we first performed a 5-fold validation to predict TCRs interacting with epitopes already present in the training set. The pan-epitope predictor demonstrated performances similar or lower to the pMHC-specific MixTCRpred predictor (Fig. 3A and Supplementary Fig. 6A). The lower prediction accuracy of the pan-epitope model was especially significant for epitopes with more than 50 TCRs (Fig. 3B and Supplementary Fig. 6B). Overall, this indicates that incorporating all available TCR–epitope pairs in the training of MixTCRpred is less effective for TCR–epitope predictions than training specific models for each epitope, thereby supporting our choice of an epitope-specific architecture in the final version of MixTCRpred. These results are consistent with previous studies[35].

Next, we investigated the ability of the pan-epitope model to predict TCRs interacting with unseen epitopes by performing a leave-one-epitope-out validation. This analysis revealed limited accuracies, with a median AUC of 0.59 (Fig. 3C). Out of 16 cases with AUCs > 0.8, 11 of them had an epitope in the training set differing by only one amino acid (Fig. 3D). We next computed the sequence similarity between each pair of epitopes restricting to epitopes presented by the same MHC allele (similarity of 1 corresponds to identical epitopes, see "Methods"). When the test epitope in the leave-one-epitope-out validation had high sequence similarity with one of the epitopes in the training set, the pan-epitope predictor gives, in general, better than random predictions (Fig. 3D). As the similarity between the test epitope and those in the training set decreases, predictions become close to random (Fig. 3D). Overall, this suggests a model where predictions can be transferred to a new epitope almost only when highly similar epitopes restricted to the same MHC are part of the training set and have enough experimentally determined TCRs interacting with them. As an example for this observation, TCRs binding to HLA-A*02:01,ELAGIGILTV exhibited similar motifs to those binding to HLA-A*02:01,E**A**AGIGILTV - the two epitopes differing only at the unexposed HLA anchor position (BLOSUM similarity of 0.89) - whereas the similarity was lower with more different epitopes, such as HLA-A*02:01, **KLVAL**G**INA**V (BLOSUM similarity of 0.39) (Fig. 3E). Including ELAGIGILTV TCR sequences in the model could be informative to predict E**A**AGIGILTV TCRs, but not for **KLVAL**G**INA**V. To evaluate the likelihood of a given epitope showing high enough similarity to one epitope in the training set of MixTCRpred, we collected all T-cell epitopes in IEDB[39]. We observed that less than 0.03% have sequence similarity higher than 0.8 (Supplementary Fig. 7).

Overall, our results show that extending predictions to unseen epitopes is challenging with the current amount of TCR–pMHC sequence data, with successful predictions possible only when the unseen epitope has very high sequence similarity and the same MHC restriction with at least one epitope in the training set of MixTCRpred. These findings align with observations from previous studies[42,65,66], further supporting the fact that generalizing predictions to any epitope is currently an unmet challenge, irrespective of the architecture of the algorithm that is used.

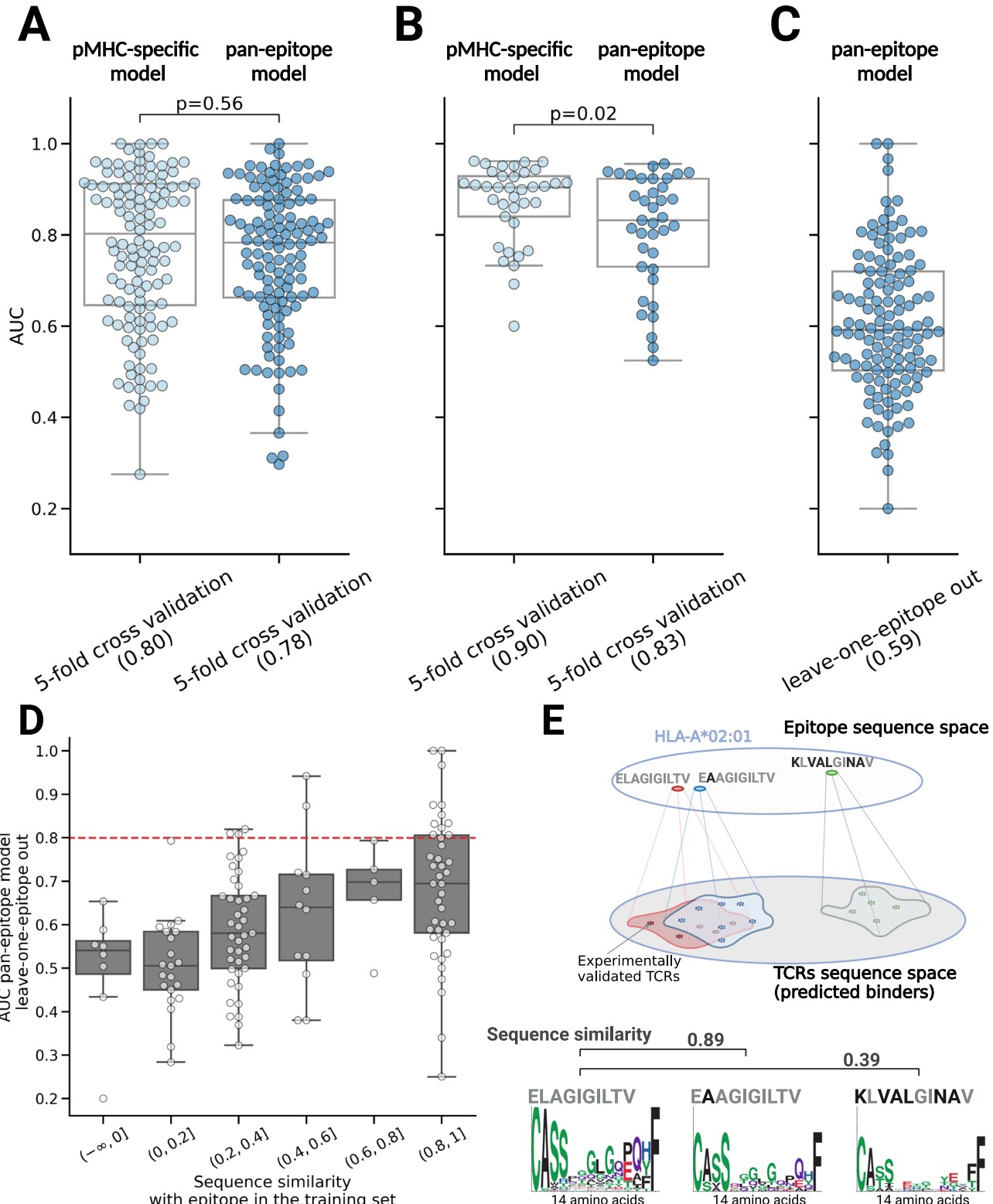

**Fig. 3 | MixTCRpred reveals how much predictions can be extended to unseen epitopes. A** Comparison of the 5-fold cross-validation average AUCs to predict TCRs for epitopes present in the training set. We used the pMHC-specific and the pan-epitope version of MixTCRpred, and included a total of $N = 120$ epitopes in the analysis. **B** Comparison of the 5-fold cross-validation average AUCs for pMHC-specific and the pan-epitope version of MixTCRpred considering $N = 37$ epitopes with more than 50 TCRs. The $p$-values were obtained with a two-sided paired $t$-test. **C** AUC values of the leave-one-epitope out validation for $N = 120$ epitopes achieved with the pan-epitope version of MixTCRpred. **D** Sequence similarity between the test epitopes and the most similar epitope in the training set, and the AUC values of the leave-one-epitope out validation for a total of $N = 120$ epitopes. The red dashed line corresponds to an AUC of 0.8. **E** Illustration for the extrapolation of TCR−epitope interaction predictions to unseen epitopes. Epitopes with very similar sequences are expected to be recognized by similar TCRs displaying similar sequence motifs. The center line within the box represents the median value, with the bottom and top bounds of the box delineating the 25th and 75th percentiles, and whiskers are set to 1.5 times the interquartile range. Source data are provided as a Source Data file.

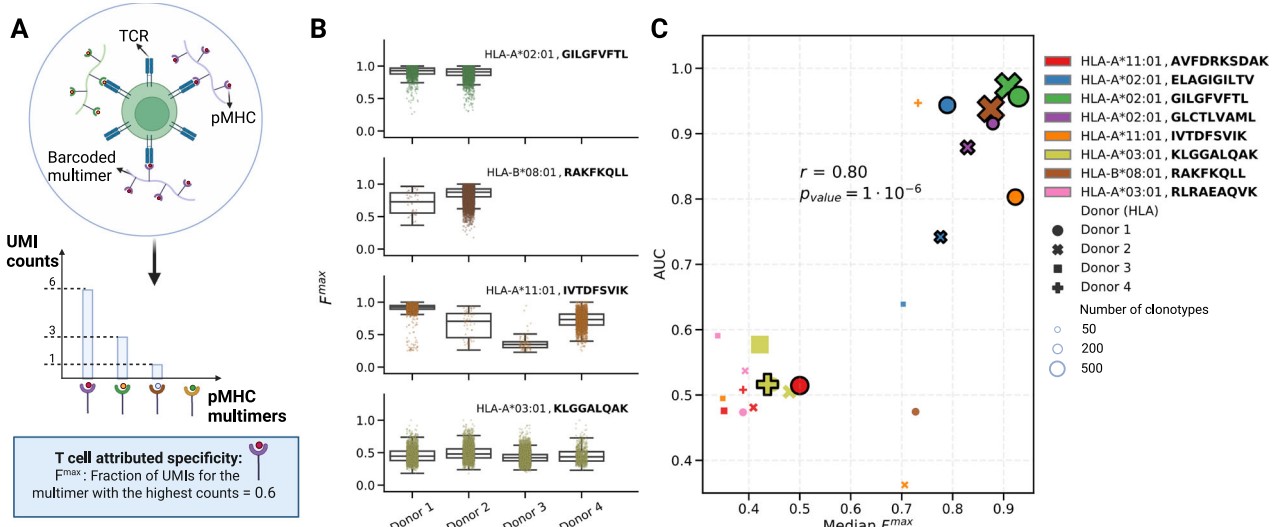

**Fig. 4 | MixTCRpred provides a quality control tool for scTCR-Seq data of T cells labeled with DNA-barcoded pMHC multimers. A** Illustration of a T cell labeled with DNA-barcoded pMHC multimers, together with the distribution of UMI for different pMHC multimers and the corresponding $F^{max}$. **B** $F^{max}$ values for each T cell in each donor-pMHC sample. The center line within the box represents the median value, with the bottom and top bounds of the box delineating the 25th and 75th percentiles, respectively. Whiskers are set to 1.5 times the interquartile range.

**C** Comparison between the median $F^{max}$ in each donor-pMHC sample and the AUC of the corresponding MixTCRpred model. The size of each point is proportional to the number of clonotypes. The black border indicates matches between the donor MHC alleles and the MHC of the multimer. The Pearson correlation and the corresponding two-sided p-value are reported. Source data are provided as a Source Data file.

## MixTCRpred provides a quality control tool for scTCR-Seq data of T cells labeled with DNA-barcoded pMHC multimers

DNA-barcoded pMHC multimers provide a powerful way to simultaneously label T cells recognizing distinct epitopes. This approach was recently used by 10x Genomics to identify and sequence T cells specific for 44 different epitopes[38]. In this assay, the binding specificity of a T cell was determined by the DNA-barcoded pMHC multimer with the highest UMI counts across all possible multimers (Fig. 4A). However, TCR–epitope interaction predictors trained on this dataset, in general, achieved poor performances[35,40]. To shed light on this issue, we calculated for each cell the fraction of UMIs specific to the pMHC multimer with the highest counts, hereinafter $F^{max}$ (Fig. 4A and "Methods"). This revealed high variability of $F^{max}$ distributions across donors and epitopes (Fig. 4B). Next, to mimic the situation where pMHCs multimers without previously known interacting TCRs are being used in individual donors, we trained and tested a specific MixTCRpred model for each combination of pMHCs and donors, and compared the AUC obtained from standard 5-fold cross-validation with the median $F^{max}$. A clear correlation between these two values was observed, indicating that cases (i.e., donor–pMHCs) with multiple pMHC barcodes per cell demonstrated low internal consistency in their TCR sequences (Fig. 4C). These include cases with a large training set (e.g., HLA-A*03:01, KLGGALQAK, with 19753 specific T cells corresponding to 7182 different clonotypes for the 4 donors, Fig. 4B, C). On the contrary, when the T cell specificity was unambiguous, reflected by a high fraction of UMIs for a specific pMHC multimer (e.g., the HLA*A-02:01, GILGFVFTL specific T cells), accurate predictions could be achieved, indicating high-quality training data. These observations motivated us to only include data from donor-pMHC with a median $F^{max} > 0.75$ (i.e., 1704 TCR clonotypes specific for 5 pMHCs, see "Methods") to train MixTCRpred.

For most cases with high MixTCRpred AUC, we observed that the MHC alleles used in the multimers were also found in the corresponding donors (points with a black border in Fig. 4C, donor HLAs in Supplementary Table 1). One exception consists of donor 4, where the IVTDFSVIK epitope in complex with HLA-A*11:01 was used. This donor was HLA-A*03:01 positive, and the two alleles show highly similar

motifs (Supplementary Fig. 8), suggesting that TCRs isolated with the HLA-A*11:01 multimer may be cross-reactive with the same epitope in complex with HLA-A*03:01.

Overall, our findings show that MixTCRpred is a valuable quality control tool for single-cell TCR-Seq data of epitope-specific T cells labeled with barcoded pMHC multimers in different donors.

## MixTCRpred reveals epitope-specific chains in epitope-specific dual α T cells

Approximately 10% of T cells express two distinct α chains on the cell surface[24]. Many approaches assume that the chain with higher expression (higher UMI counts or higher read counts if both chains have the same UMI count) is the one mediating epitope recognition[27,29]. To investigate the validity of this assumption, we focused on the 10x Genomics dataset[38] and selected two pMHCs with a large number of αβTCRs, namely HLA-A*02:01, GILGFVFTL (839 αβTCRs) and HLA-A*02:01, ELAGIGILTV (169 αβTCRs). Next, we retrieved T cells expressing two α and one β chains (152 for HLA-A*02:01, GILGFVFTL, and 18 for HLA-A*02:01, ELAGIGILTV) after filtering out doublets (see "Methods").

For each epitope, we trained a specific MixTCRpred model with the αβTCR sequences from single α T cells. We then used it to predict the binding of TCRs from dual α T cells ($\alpha_x\alpha_y$-β), by considering each α chain separately ($\alpha_x$–β and $\alpha_y$–β TCRs, see Fig. 5A). In most cases, the two α chains exhibited significantly different MixTCRpred scores (Fig. 5B). The best predicted MixTCRpred binder coincided with the α chain with higher expression in approximately 60% of the dual α T cells (85 cases out of 152 for the HLA-A*02:01, GILGFVFTL, and 11 out of 18 for the HLA-A*02:01, ELAGIGILTV) (Supplementary Fig. 9A). To validate that the α chain with the best MixTCRpred prediction (hereinafter chain $\alpha_1$) was the one involved in epitope recognition, we selected a set of 6 dual α T cells for HLA-A*02:01, GILGFVFTL and 5 for HLA-A*02:01, ELAGIGILTV (Tables 1 and 2 and Supplementary Fig. 10). RNA encoding $\alpha_1$βTCR and $\alpha_2$βTCR from these dual α T cells was synthesized and electroporated into TCR–Jurkat cells. After overnight incubation, TCR transfected cells were interrogated by pMHC-multimer staining (see "Methods"). Figure 5C demonstrates that α chains predicted by

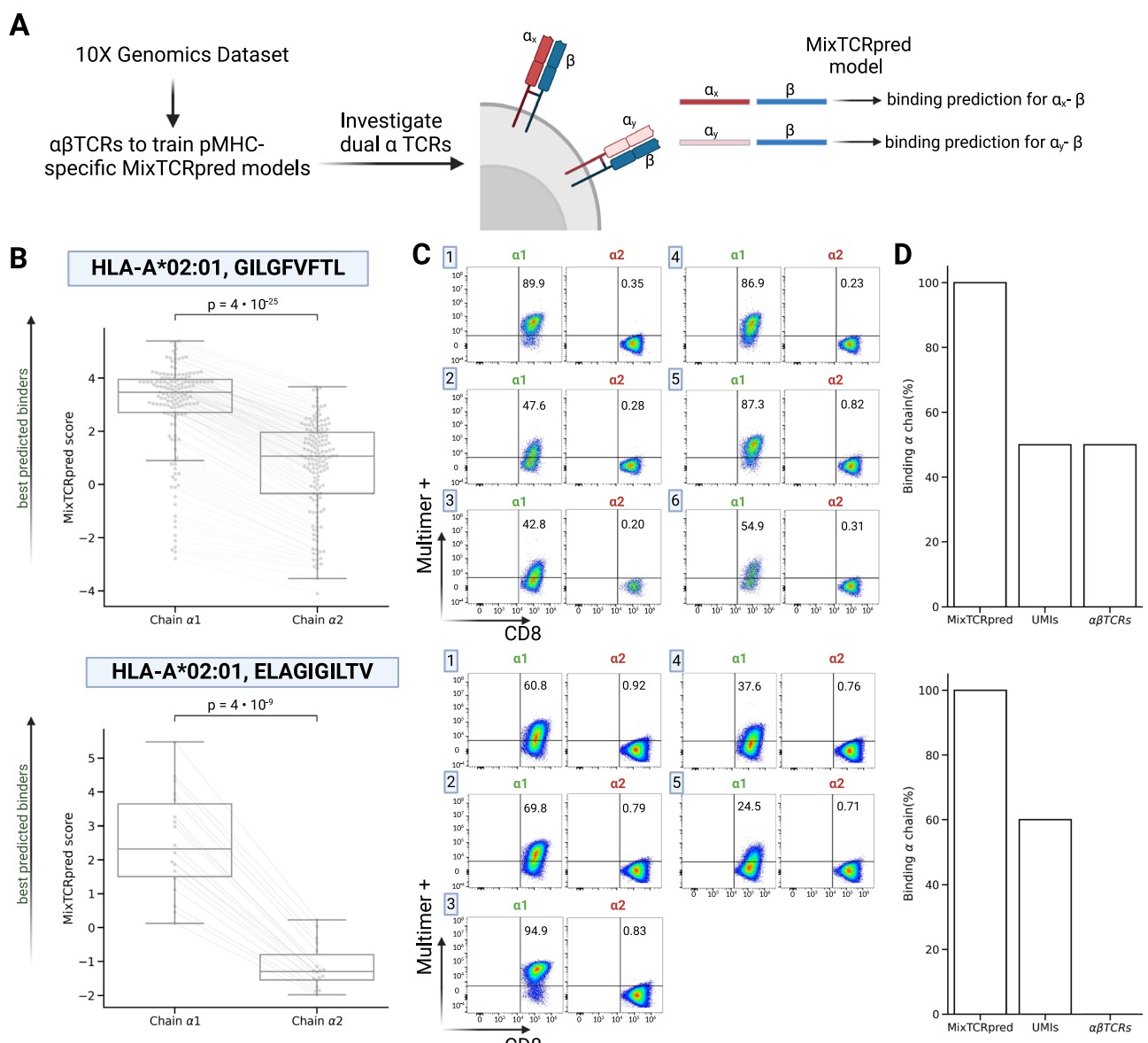

**Fig. 5 | MixTCRpred reveals epitope-specific chains in epitope-specific dual α T cells. A** Overview of our pipeline to investigate epitope-specific chains in dual α T cells. **B** MixTCRpred scores of the two TCRs in dual α T cells specific to HLA-A*02:01, GILGFVFTL (N = 152) and to HLA-A*02:01, ELAGIGILTV (N = 18). Chain α1 is defined as the one with the best MixTCRpred score. The center line within the box represents the median value, with the bottom and top bounds of the box delineating the 25th and 75th percentiles, respectively. Whiskers are set to 1.5 times the interquartile range. The p-values were obtained with a two-sided independent t-test. **C** Multimer staining of the TCRs with the predicted α chain (chain α1) and TCRs with the other α chain (chain α2) for the dual α T cells used in our experimental validation. **D** Fraction of correctly predicted α chains by MixTCRpred, considering the most expressed chains, and with exact matches in αβTCRs from single α T cells. Source data are provided as a Source Data file.

**Table 1 | List of the six dual α TCRs specific to the HLA-A*02:01, GILGFVFTL epitope, which were selected for experimental validation**

|   | TRBV | CDR3 TRB | TRBJ | TRAV (α1) | CDR3 TRA (α1) | TRAJ (α1) | TRAV (α2) | CDR3 TRA (α2) | TRAJ (α2) |
|---|------|----------|------|-----------|---------------|-----------|-----------|---------------|-----------|
| 1 | TRBV19 | CASSIGSYGYTF | TRBJ1-2 | TRAV38-1 | CAFMISAGGTSYGKLTF | TRAJ52 | TRAV12-2 | CAVIGGGADGLTF | TRAJ45 |
| 2 | TRBV19 | CASSIRSSYEQYF | TRBJ2-7 | TRAV12-2 | CAVNQGGGSQGNLIF | TRAJ42 | TRAV30 | CGTEWEARLMF | TRAJ31 |
| 3 | TRBV19 | CASSTGVYGYTF | TRBJ1-2 | TRAV38-1 | CAFMTNAGGTSYGKLTF | TRAJ52 | TRAV30 | CGTERSGGSNYKLTF | TRAJ53 |
| 4 | TRBV19 | CASSIGLYGYTF | TRBJ1-2 | TRAV38-2DV8 | CAYSVNAGGTSYGKLTF | TRAJ52 | TRAV40 | CLLEVFFGNEKLTF | TRAJ48 |
| 5 | TRBV19 | CASSSRAGGEQYF | TRBJ2-7 | TRAV5 | CAENEGGGSQGNLIF | TRAJ42 | TRAV30 | CGTRKNDYKLSF | TRAJ20 |
| 6 | TRBV19 | CASSQGSWGYTF | TRBJ1-2 | TRAV38-1 | CAFMIGAGGTSYGKLTF | TRAJ52 | TRAV20 | CAVFFEGGATNKLIF | TRAJ32 |

The α1 chain is the best MixTCRpred prediction.

**Table 2 | List of the five dual α TCRs specific to the HLA-A\*02:01, ELAGIGILTV epitope, which were selected for experimental validation**

| | TRBV | CDR3 TRB | TRBJ | TRAV (α1) | CDR3 TRA (α1) | TRAJ (α1) | TRAV (α2) | CDR3 TRA (α2) | TRAJ (α2) |
|---|---|---|---|---|---|---|---|---|---|
| 1 | TRBV6-3 | CASTLGEGSEAFF | TRBJ1-1 | TRAV12-2 | CRVGGGADGLTF | TRAJ45 | TRAV21 | CDRGRGTSYDKVIF | TRAJ50 |
| 2 | TRBV4-2 | CASSQGAFSVEQYF | TRBJ2-7 | TRAV12-2 | CAVKGGGADGLTF | TRAJ45 | TRAV14DV4 | CAMSISYNNNDMRF | TRAJ43 |
| 3 | TRBV6-1 | CASSDTETGGLETQYF | TRBJ2-5 | TRAV12-2 | CAVNGARLMF | TRAJ31 | TRAV5 | CAETTGALYSGAGSYQLTF | TRAJ28 |
| 4 | TRBV5-8 | CASSFGALNTEAFF | TRBJ1-1 | TRAV12-2 | CAVCSGGYNKLIF | TRAJ4 | TRAV8-2 | CDRYSTLTF | TRAJ11 |
| 5 | TRBV14 | CASSFQGLGTEAFF | TRBJ1-1 | TRAV12-2 | CAVNNAGGTSYGKLTF | TRAJ52 | TRAV13-2 | CAEKDDKIIF | TRAJ30 |

The α1 chain is the best MixTCRpred prediction.

MixTCRpred were binding to the pMHCs in all cases, while the other chains did not bind. Similar results were not obtained with predictions based on the highest UMI count, which identified the correct epitope-specific α chains only in 50% and 60% of the tested cases (Fig. 5D and Supplementary Fig. 11).

An alternative approach to identify epitope specific α chains is to look for exact matches of the $α_x$-β or $α_y$-β TCRs in our comprehensive database of single α T cells. Exact matches could be found for 96 cases for HLA-A\*02:01, GILGFVFTL, including only 3 cases (TCR-1, -4, and -5) for the TCR sequences experimentally tested, and none for the HLA-A\*02:01, ELAGIGILTV epitope (Fig. 5D and Supplementary Fig. 8B).

Overall, these results indicate that MixTCRpred offers a robust framework for identifying epitope-specific chains in dual α T cells and overcomes limitations of methods such as UMI counts, or exact TCR sequence matches in single α T cells.

**MixTCRpred reveals enrichment of TCRs specific for an immunodominant SARS-CoV-2 epitope in COVID-19-positive patients**
αβTCR repertoires have been sequenced in several COVID-19 patients to characterize the T cell response to SARS-CoV-2 infection, but the precise epitope targets have largely remained unknown. Earlier investigations have identified T cells specific to SARS-CoV-2 by examining TCRs that are enriched in COVID-19-positive patients with respect to healthy patients[67], or that are oveshared between COVID-19-positive patients[68]. Cross-referencing the COVID-enriched TCRs with a bulk TCR dataset with known specificity for certain SARS-CoV-2 peptide[69] also enabled the identification of epitope targets for a substantial number of clonotypes[67].

As a complementary approach, here we utilized MixTCRpred to explore the presence of TCRs specific to a known immunodominant class II SARS-CoV-2 epitope (TFEYVSQPFLMDLE from the SARS-CoV-2 spike protein and restricted to HLA-DPB1\*04:01) in TCR repertoires. We collected αβTCR repertoires of CD4⁺ T cells from multiple studies that isolated T cells from the peripheral blood of both COVID-19-positive and COVID-19-negative patients[70–72]. This collection included T cells that were stimulated with a range of SARS-CoV-2 proteins[70,71], as well as T cells that were sequenced directly ex-vivo without any stimulation[72], with a total of 205,930 CD4⁺ T cells from 138 COVID-19-positive patients and 46 healthy donors (see "Methods"). The HLA alleles of the patients were not provided.

Next, we calculated the proportion of TCRs that were predicted to target the TFEYVSQPFLMDLE epitope within each TCR repertoire. A threshold of 0.01 was used on MixTCRpred %rank (see "Methods" and Supplementary Fig. 1). Across all three studies, we observed an enrichment and overall higher fraction of CD4⁺ T cells predicted to target this immunodominant epitope in repertoires from COVID-19-positive patients (Fig. 6A, B). Among our predictions, we also observed several cases of expanded CD4⁺ T cells (Fig. 6C). The overall ratio of T cells predicted to be TFEYVSQPFLMDLE specific was particularly pronounced in samples from the Bacher et al. study[70]. In this study CD4⁺ T cells were stimulated with peptides from the SARS-CoV-2 spike protein that included the TFEYVSQPFLMDLE peptide. Conversely, in

the Meckiff et al. study[71], a different peptide pool was used for stimulation, which did not encompass the TFEYVSQPFLMDLE peptide[73], and the overall enrichment was less prominent. The expanded clones were less frequent in unstimulated cells from PBMC, as expected for epitope-specific CD4⁺ T cells. Our results indicate that MixTCRpred offers a robust framework for in silico analysis of epitope-specific T cells directly from TCR repertoires and reveals enrichment of T cells predicted to be specific for the immunodominant DPB1\*04:01,TFEYVSQPFLMDL epitope in COVID-19-positive patients.

## Discussion
TCR sequencing enables researchers to rapidly determine the TCR repertoire of clinically relevant samples like tumors. One of the main promises of TCR–pMHC interaction predictors is the ability to identify in silico TCRs recognizing specific epitopes directly from such TCR repertoire data. In this work, we trained a predictor of TCR–pMHC interactions for a set of epitopes with enough known TCRs. These epitopes cover many common viruses, as well as some cancer antigens.

Our work indicates that reasonable prediction accuracy requires at least 50 αβTCRs. These results provide a strong motivation for global initiatives to collect many TCRs recognizing diverse epitopes, and we anticipate that both the data collected and curated in this work and the MixTCRpred framework will contribute to this global endeavor.

Several existing tools have attempted to extrapolate predictions to other epitopes, including epitopes without any known TCRs (i.e., unseen epitopes), by considering both the TCR and the epitope sequences in the input of their machine learning framework. While some approaches have reported some success[47], other studies have reached the conclusion that extending predictions to any epitope is currently not feasible[65,66]. Our results, which align with previous findings[45,46,74], suggest that cases where predictions could be extrapolated to new epitopes consist mostly of epitopes having very high similarity and the same MHC restriction to an epitope in the training set. As of today, these cases represent only a tiny minority of all known epitopes. Moreover, we cannot fully exclude some level of circularity since TCRs tested with a given epitope may have been selected based on their similarity with TCRs interacting with another highly similar epitope. These observations indicate that extrapolation to any epitope is still an unsolved challenge. Our work also suggests that mixing TCRs interacting with different epitopes when training TCR–pMHC interaction predictors, in general, does not improve and may even lower the prediction accuracy, thereby justifying our choice to train a separate predictor for each epitope.

Application of MixTCRpred to data generated with the 10x Genomics immune profiling platform[38] suggests that several cases (i.e., pMHC–donor pairs) may include a substantial fraction of contaminants. This observation has important consequences since 10x Genomics data currently constitute 70% of all paired αβTCR–epitopes in VDJdb[28] and 66% in IEDB[39]. Filtering out putative contaminants led to the removal of roughly 85% of the 10x Genomics data, which is consistent with estimates of contaminants in other studies[27,40]. Some of

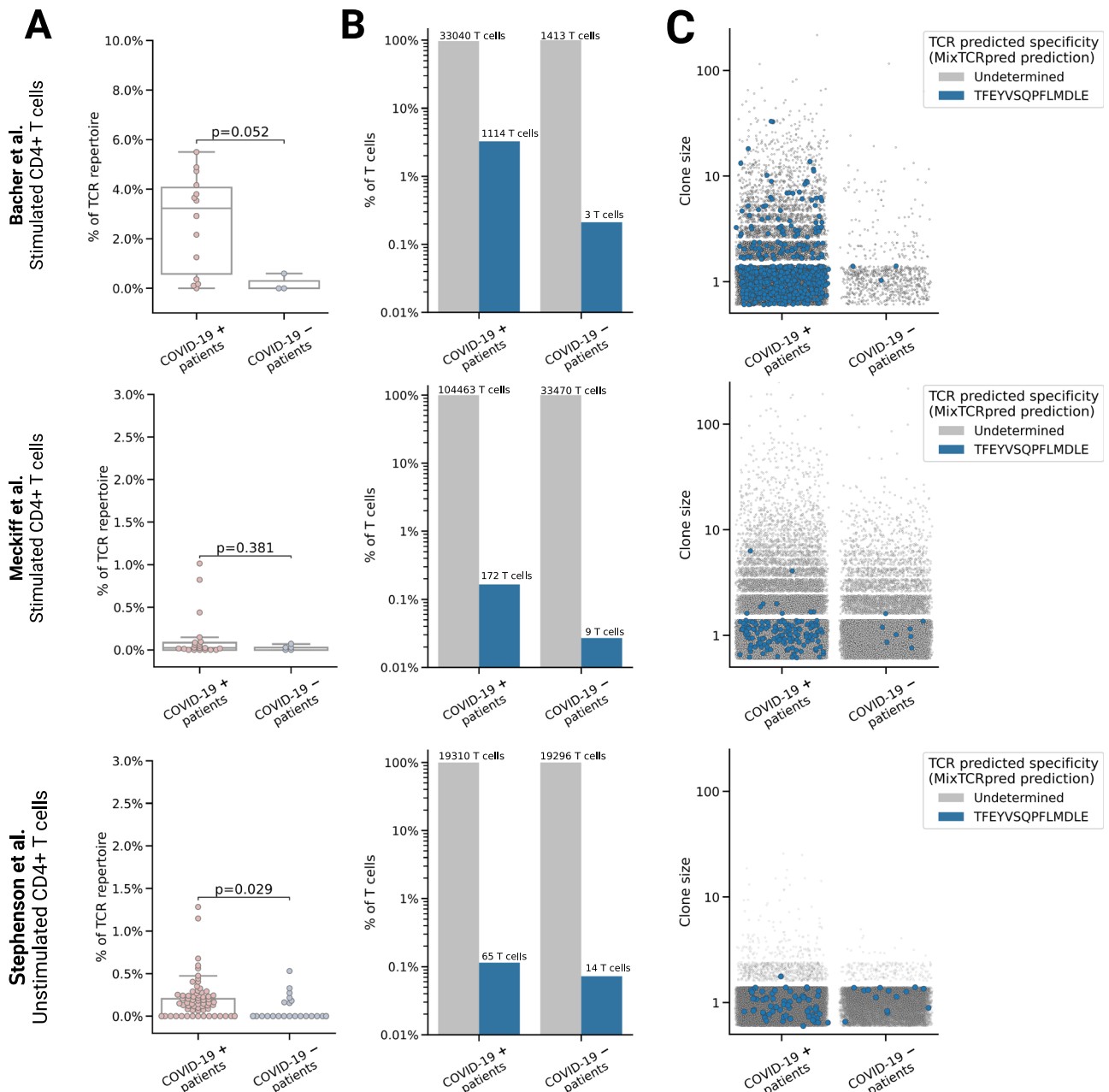

**Fig. 6 | MixTCRpred reveals enrichment of TCRs specific for an immunodo-minant SARS-CoV-2 epitope in COVID-19-positive patients.** TCR repertoires of stimulated CD4[+] from Bacher et al.[70] (14 COVID-19-positive and 3 COVID-19-negative patients) Meckiff et al.[71] (21 positive and 5 negative patients), and of unstimulated CD4[+] from Stephenson et al.[72]. **A** Fraction of T cells predicted to be TFEYVSQPFLMDLE specific for each patient. The center line within the box represents the median value, with the bottom and top bounds of the box delineating the 25th and 75th percentiles, respectively. Whiskers are set to 1.5 times the interquartile range. The *p*-values were obtained with a two-sided independent *t*-test. **B** Overall fraction of T cells with undetermined specificity (in gray) or pre-dicted to be specific for the immunodominant TFEYVSQPFLMDLE epitope (in blue). **C** Clone size of each TCR clonotype. Source data are provided as a Source Data file.

these studies include HLA mismatch between the donor and the pMHC multimers in the filtering criteria[40]. Our results support this approach but suggest integrating the notion of HLA binding motif divergence and keeping cases of mismatched HLAs with similar binding motifs (e.g., HLA-A*03:01 and HLA-A*11:01). The remaining TCR–pMHCs pairs demonstrate consistent TCR sequence patterns and good internal AUC. These data were highly valuable for improving and expanding the epitope coverage of MixTCRpred. We anticipate that improvements in both DNA barcoded multimer technology and post-processing tools will enable researchers to collect large amounts of TCR–pMHCs interactions in the near future with this technology. Such data will be

instrumental in characterizing the TCR specificity of viral or cancer epitopes and possibly one day for training TCR–pMHC interaction predictors for any epitope.

Our work shows that MixTCRpred can accurately identify the epitope-specific chain in dual α T cells recognizing specific epitopes, even when this chain did not have the highest UMI count and was not present among the single α chain T cells. These results have important implications for the processing of single-cell TCR-sequencing data from epitope-specific T cells. For instance, several entries in the VDJdb[29] report the α chain with the highest UMI count, while the MixTCRpred score is much higher for the other α chain. These include

the α sequences TRAV30, CGTEWEARLMF, TRAJ31 and TRAV20, CAVFFEGGATNKLIF, TRAJ32 for the HLA-A*02:01 GILGFVFTL epitope, as well as TRAV5, CAETTGALYSGAGSYQLTF, TRAJ28 and TRAV13-2, CAEKDDKIIF, TRAJ30 for HLA-A*02:01 ELAGIGILTV epitope, which we experimentally validated as non-binders (Tables 1 and 2 and Source Data file). TCR–pMHC interaction prediction tools like MixTCRpred will help address this issue and improve the quality of data stored in these databases.

The clear differences observed between the scores of the two chains in dual α T cells also indicate that most of these cases were not doublets and that, in general, only one α chain is responsible for the epitope specificity and cases where both α chains recognize the same epitope appear to be very rare. Our observations further suggest that the highest UMI criteria should be used with caution, also when dealing with dual α T cells of unknown specificity.

The enrichment and expansion of TCRs predicted to target an immunodominant SARS-Cov-2 epitope in TCR repertoires from COVID-19-positive patients, even without the possibility of stratifying patients based on their HLA alleles, suggests that many of these clonotypes are specific for TFEYVSQPFLMDLE SARS-CoV-2 epitope. This is consistent with the fact that this epitope is immunodominant[73] and restricted to a frequent HLA-DP allele (i.e., HLA-DPB1*04:01, found in >40% in diverse populations[75]). Moreover, the HLA-DPB1*04:01 motif has high similarity with many other HLA-DP alleles (Supplementary Fig. 12). TCRs predicted to target TFEYVSQPFLMDLE in COVID-19-negative donors or in COVD-19-positive donors with incompatible alleles may represent cross-reactive TCRs with other epitopes, as expected from previous observations of SARS-CoV-2 reactive TCRs in patient before COVID-19 infection[76,77].

In summary, our study provides a high-quality dataset of TCR–pMHC interactions for several common viral and cancer epitopes (reported in the Source Data file) as well as a robust command-line tool (https://github.com/GfellerLab/MixTCRpred) to predict new TCRs binding to these epitopes. Beyond computational annotation of TCR repertoires, our work shows that MixTCRpred can be used as a quality control tool for single-cell TCR-sequencing data of T cells labeled with DNA barcoded multimers, as well as to annotate α chains mediating epitope recognition in epitope-specific dual α T cells. Considering the rapid developments of technologies to isolate and sequence epitope-specific T cells, we anticipate that the epitope coverage of TCR–pMHC interaction predictors will keep increasing, making such tools relevant for in silico identification of TCRs recognizing known viral or cancer epitopes directly from TCR repertoire. This could pave the way for diagnosis applications, as illustrated by the SARS-CoV-2 epitope analyzed in this work.

## Methods

### TCR–epitope sequence data
TCR–pMHC pairs were collected from publicly available datasets, including VDJdb[29], (data download 27/10/2022), IEDB[39] (data download 02/11/2022), and the McPAS database[60] (data download 27/10/2022). TCR–pMHCs pairs from the 10x Genomics dataset[38] were processed separately. Additional data for *Mus musculus* were retrieved from two recent studies[61,62]. Only paired TCR sequences (with both the α and the beta β sequences) were considered, and sequences containing non-standard amino acids were removed. Duplicated TCR–pMHC were merged based on V/J gene usage and CDR3 sequence for both the α- and β-chain. The data analysis was done with Python (v.3.9.7) using the BioPython (v.1.79) and pandas (v.1.5.2) libraries.

### Pre-processing single-cell dataset
Multiple datasets used in this study were generated using the Chromium platform of 10x Genomics[38,61,62,78] and processed with the Cell Ranger Single Cell Software Suite by 10x Genomics. To ensure high-quality data standard quality control on transcriptomic data was performed, by

a. Filtering out cells with low/high UMIs (<1500 or >15,000 UMIs and remove top/bottom 1%)
b. Filtering out cells with a low number of genes (<700 UMIs and removing top/bottom 1%)
c. Filtering out cells with high mitochondrial/ribosomal data (<10% mitochondrial gene, <50% ribosomal genes)

The analysis was done with the scanpy library[79] The scirpy library[80] (v.0.10.0) was used to integrate TCR sequence with transcriptomics data and to identify T cells with multiple chains. Doublets were identified and removed with the scrublet package[81].

### 10x Genomics dataset
For the 10x Genomics dataset[38], after standard single-cell dataset preprocessing, one additional step is required to match each TCR with the cognate DNA-barcoded multimer. Following the guidelines outlined in the 10x Genomics documentation, cells with less than 10 multimer UMI counts were filtered out. Additionally, cells were also removed if their UMI counts for a specific multimer were not significantly higher than the UMI counts for negative control multimers (at least 5 times greater than the negative controls). Finally, cells were excluded if they had UMI counts for more than 5 different multimers. Each remaining cell was then matched to the multimer with the highest UMI counts, which was attributed to the specificity of the T cell (a total of 67,084 epitope-specific T cells and 14,887 αβTCR clonotypes). 50,625 of them were αβ T cells (10,376 clonotypes), but only 26,753 T cells (1704 clonotypes) had a median fraction of UMI counts for one specific multimer >0.75 and were thus included in the MixTCRpred training dataset.

### Negative data
For each epitope, negative cases (i.e., TCRs not binding to the target epitope) were computationally generated by

- sampling TCRs specific to other pMHCs (negative/positive ratio of 1:1). In order to avoid the model from learning biased patterns due to the imbalanced distribution of TCR–pMHCs sequence data (Fig. 1C, D), a weight was assigned to each sequence before sampling. This weight was calculated as the reciprocal of the total number of TCRs that bind to the corresponding epitope. *Homo sapiens* and *Mus musculus* epitopes were treated separately.
- sampling TCRs from TCR repertoires (negative/positive ratio of 4:1). *Homo sapiens* αβTCR repertoires were downloaded from iReceptor[9], and from one recent study for *Mus musculus*[78]. When studying specifically the 10x Genomics dataset (leave-one-sample out, MixTCRpred for quality control of the 10x Genomics dataset, MixTCRpred to investigate dual α T cells), TCRs not assigned to any epitopes (UMIS counts = 0) were used as negatives[35,36].

As a result, the final dataset had a negative-to-positive ratio of 5:1.

### MixTCRpred model
MixTCRpred is a transformer-based model[64] written in Python, relying on the PyTorch[82] and PyTorchLighting[83] libraries. For each pMHC in our dataset a specific MixTCRpred model was trained with experimentally validated TCRs and computationally generated negatives. For each TCR used as input, CDR1, CDR2 (from the V gene), and CDR3 sequences for the α- and the β-chain were retrieved separately. The sequences were padded, concatenated, and numerically embedded using the nn.Embedding function of PyTorch (learned embedding) and a positional encoding. A transformer encoder was then used[64], followed by a dense classification layer to output the MixTCRpred binding score (with higher score sequences more likely to

bind). To achieve comparability across models, for each input TCRs the corresponding % rank was also calculated. To this end, *Homo sapiens* αTCRs and βTCRs from iReceptor[9] were collected. For *Mus musculus* αTCRs were downloaded iReceptor[9] while βTCRs from three different studies[84–86] from the immuneACCESS website. Next, treating *Homo sapiens* and *Mus musculus* separately, α and β TCR sequences were randomly paired to generate $10^6$ different TCRs that were scored using each one of the 146pMHC MixTCRpred models. The %rank of an input TCR with a given MixTCRpred score is the fraction of TCR sequences with higher scores multiplied by 100. Low %rank score sequences are more likely to bind. Most true binders have been observed to have a % rank of less than 0.5–0.1 (Supplementary Fig. 1).

### Benchmarking with other existing tools
To benchmark MixTCRpred with other pre-trained tools accessible through command-line or web interfaces, the entire dataset was used for training except for the McPAS dataset[60], which was kept aside as the test set. Sequences appearing in both the training and test sets were removed from the test set. A distinct MixTCRpred model was trained for each epitope having more than 50 binding TCRs in the training set and more than 10 TCR in the test set. The following pre-trained TCR–epitope interaction predictors were used in this validation:

- *ERGO2.0*[37,54] from the webserver (https://tcr2.cs.biu.ac.il/home) selecting the versions that did not include the McPAS dataset[60] in the training set. Both the Long Short-Term Memory (LSTM) and the AutoEncoders (AE) based were considered.
- *NetTCR2.1*[35] from the webserver (https://services.healthtech.dtu.dk/services/NetTCR-2.1/) which enabled predictions for six human epitopes. NetTCR2.1 included the McPAS dataset[60] in the training set.
- *tcrdist*[50] from the Python toolkit available at https://github.com/kmayerb/tcrdist3 and the corresponding tutorial for TCR–pMHC interaction prediction, which did not include the McPAS dataset in the training set.

The IMMREP22 dataset[43] released during the IMMREP22 TCR–epitope specificity workshop, was used to benchmark MixTCRpred with other predictors. The IMMREP22 dataset consisted of curated TCR sequence data for 17 pMHCs, each having a minimum of 50 unique αβTCR sequences. It was downloaded from https://github.com/viragbioinfo/IMMREP_2022_TCRSpecificity where the AUCs of prediction tools that participated in the workshop were also provided. The prediction scores for individual TCRs were not provided. ERGO2.0[37,54] was not part of this validation and was separately re-trained and tested.

### Sequence similarity
To compute the sequence similarity between a test epitope and an epitope in the training set, the two sequences were aligned with the pairwise2 align function from the BioPython package[87] using the BLOSUM62 scoring matrix[88]. The resulting pairwise alignment score was then divided by the score obtained, aligning the test epitope sequence with itself so that the maximal similarity score was 1. The similarity score can assume negative values due to negative entries in the BLOSUM62 matrix. Scores closer to 1 indicate greater similarity between peptide pairs.

### Peptides and pMHC multimers production
Peptides and HLA-A*02:01,GILGFVFTL and HLA-A*02:01,ELAGIGILTV multimers were produced by the Peptides and Tetramers Core Facility (PTCF) of the Department of Oncology, University of Lausanne and University Hospital of Lausanne. HPLC purified peptides (≥90% pure), were verified by UHPLC-MS and kept lyophilized at −80 °C.

Peptide–MHC multimers were prepared fresh and used within a week or kept aliquoted at −80 °C[89].

### TCR validation
To validate antigen specificity and interrogate T cell reactivity vs HLA-A*02:01,GILGFVFT and HLA-A*02:01,ELAGIGILTV epitopes, TCRα/β pairs were cloned into Jurkat T cells (TCR/CD3 stably transduced with human CD8α/β and TCRα/β CRISPR-KO), as previously described[77,90]. In brief, codon-optimized DNA sequences coding for paired α and β chains, including the mouse constant region instead of the human one, were synthesized at GeneArt (Thermo Fisher Scientific) or Telesis Bio DNA. The DNA fragments served as templates for in vitro transcription (IVT) and polyadenylation of RNA molecules as per the manufacturer's instructions (Thermo Fisher Scientific), followed by co-transfection into recipient T cells. Jurkat cells were electroporated using the Neon electroporation system (Thermo Fisher Scientific) with the following parameters: 1325 V, 10 ms, and three pulses. After overnight incubation, electroporated Jurkat cells were interrogated by pMHC-multimer staining with the following surface panel: anti-hCD3 APC Fire 750 (SK7, Biolegend Cat# 641415, 0.4 μL in 50 μL); anti-hCD8 FITC (SK-1 Biolegend, Cat# 344704, 0.15 μL in 50 μL); anti-hCD4 PE-CF594 (RPA-T4, BD Bioscience Cat# 562281, 0.4 μL in 50 μL); anti-mouse TCRβ-constant APC (H57-597, Thermo Fisher Scientific, Cat# 17-5961-81, 0.6uL in 50 μL); pMHC-multimer-PE (HLA-A*02:01,GILGFVFT and HLA-A*02:01,ELAGIGILTV in-house synthesized, 1 μL in 50 μL); viability dye Aqua (L34966, Thermo Fisher Scientific, 0.15 μL in 50 μL staining mix in PBS). In total, 200,000 TCR-transfected Jurkat cells were washed once in PBS, resuspended in 50 μL of FACS buffer (5 mM EDTA, 0.2% azide, 0.2% BSA in PBS) containing 1uL of multimer and incubated for 30 min at RT. Cells were washed once in PBS and resuspended in 50uL of PBS containing LIVE/DEAD dye and the antibody cocktail for cell surface staining. Cells were incubated at 4 °C for 30 min and washed twice before acquisition. FACS data were analyzed with FlowJo 10.8.1 (TreeStar). The gating strategy is illustrated in Supplementary Fig. 13.

### SARS-CoV-2 TCR repertoires
αβTCR repertoires were collected from three different studies[70–72], and each donor was annotated as COVID-19-positive or negative.

1. Ex-vivo-isolated memory CD4+ T cells from PBMCs of unexposed donors ($n = 6$) and COVID-19-positive patients ($n = 14$) stimulated with pooled SARS-CoV-2 spike, membrane, and nucleocapsid proteins[70].
2. Ex-vivo CD4+ T cells isolated from PBMCs of 40 Covid-19 patients and 13 healthy donors. CD4+ T cells were stimulated with a SARS-CoV-2 derived peptide pool, which did not include the region of the spike protein comprising the TFEYVSQPFLMDLE peptide[71].
3. Ex-vivo unstimulated T cells from PBMCs. Cell annotations based on the RNA expression of known marker genes were provided. A total of 103 CD4+ TCR repertoires from COVID-19 positive patients and 38 from healthy donors[72] were collected.

To avoid small sample bias, samples for which less than 200 TCRs were available were removed. In total, 180,341 TCRs for 136 TCR repertoires from COVID-19-positive patients and 44 from COVID-19-negative donors were collected.

### Reporting summary
Further information on research design is available in the Nature Portfolio Reporting Summary linked to this article.

## Data availability
TCR–pMHC sequence data were collected from the publicly available datasets VDJdb[29] (https://vdjdb.cdr3.net/), IEDB[39] (https://www.iedb.

org/), and the McPAS database[60] (http://friedmanlab.weizmann.ac.il/McPAS-TCR/), and the 10x Genomics dataset[38] (https://pages.10xgenomics.com/rs/446-PBO-704/images/10x_AN047_IP_A_New_Way_of_Exploring_Immunity_Digital.pdf). Additional data for *Mus musculus* were retrieved from two recent studies[61,62] (https://www.ncbi.nlm.nih.gov/geo/query/acc.cgi?acc=GSE182320 and https://www.ncbi.nlm.nih.gov/geo/query/acc.cgi?acc=GSE201730). A total of 17,715 different TCRs for 146 pMHCs were collected in this study and used to train the MixTCRpred models. The TCR–epitope sequence dataset and a list of the pre-trained MixTCRpred models are provided in the Source Data file and at https://github.com/GfellerLab/MixTCRpred. The pre-trained MixTCRpred models for the 146 pMHCs reported in the manuscript are available at https://zenodo.org/record/7930623. All other data are available in the article and its Supplementary files or from the corresponding author upon request. Source data are provided in this paper.

## Code availability

The MixTCRpred code (v.1.0)[91] is available at https://github.com/GfellerLab/MixTCRpred. The repository includes a Google Colab notebook that enables running MixTCRpred directly from a web browser.

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

## Acknowledgements

We thank Julien Racle for constructive criticism of the manuscript and for reviewing the MixTCRpred code. Additionally, we thank Aurélie Gabriel for reviewing the MixTCRpred code and Christophe Sauvage and Alexandra Michel for their technical help. This project has received funding from the SNF Sinergia program (CRSII5_193749) to D.G and G.C, and the European Union's Horizon 2020 research and innovation program under the Marie Skłodowska-Curie grant agreement, No. 101027973 to G.C. Figures 1–6 were created with Biorender.com.

## Author contributions

D.G. and G.C. designed the study. G.C. developed the bioinformatics tools. G.C. performed the bioinformatics analyses. G.C. and D.M. performed the SARS-CoV-2 TCR repertoires analyses. S.B., J.S., and P.G. performed the experiments. A.H. supervised the experiments. G.C. wrote the paper. G.C., S.B., D.M., A.H. and D.G. provided materials and feedback on the paper.

## Competing interests

David Gfeller is a consultant for CeCaVa. The remaining authors declare no competing interests.
