## [Peer Review File · Nature Communications]

Deep learning predictions of TCR-epitope interactions reveal epitope-specific chains in dual alpha T cellsEditorial Note: This manuscript has been previously reviewed at another journal that is not operating a transparent peer review scheme. This document only contains reviewer comments and rebuttal letters for versions considered at *Nature Communications*.

REVIEWERS' COMMENTS

Reviewer #1 (Remarks to the Author):

The reviewers have addressed my previous concern by clarifying the TCR specificity prediction application scope. I think the conclusion will be helpful for readers to decide whether or not to use computational prediction methods. I would like to see this paper published.

Reviewer #4 (Remarks to the Author):

I appreciate the authors' detailed responses to Reviewer 2's comments. The manuscript is clear with Nature Communications being an appropriate venue for publication. A few of Reviewer 2's comments center around the fact that MixTCRpred works the best for only specific epitopes rather than unseen epitopes, and I think some changes will further improve the manuscript. I base my comments on the numbering of the Reviewer 2's comments and the corresponding responses.

- [Major Comment 2 and 6] The authors should more prominently make clear in the abstract that MixTCRpred works with only specifically-trained epitopes. In the current version, it is mentioned in the last sentence, but given the authors' analyses on the pan-epitope network and the purposes of MixTCRpred, this should be more clearly defined.
- [Major Comment 8] In the same vein, if say practitioners were to have different epitopes than those trained, is it easy for them to train a version of MixTCRpred for their own use? If so, this is a great asset and maybe a tutorial on the software page will be beneficial.
- [Major Comment 4] It may be better to mention this reasoning in the manuscript to discuss the choice of using the embedding layer as part of the network architecture.

Reviewer #5 (Remarks to the Author):

The author introduced an important area of research, a vaccine design by machine learning. MHC interacting class I and class II allele diversity is absent as it is of utmost importance. Further epitopes variations should be introduced CD4 and CD8. While machine learning predicted with high accuracy there is a concern of overfit problem noted that reflects higher predictability. Importantly, TCR-pMHC recognition is highly dynamic, and machine learning alone cannot predict accurately; therefore, structural parameters should be included.

REVIEWERS' COMMENTS

We thank the reviewers for their comments, which have helped to improve our manuscript. Below, the original reports are reproduced in black, and our replies in blue.

In addition to the reviewers' comment we also did the following modifications:

- a. In Figure 5b, we opted to display the MixTCRpred score instead of the MixTCRpred %rank. This adjustment does not alter the conclusion (only one of the two alpha chains is expected to bind to the target epitope), while improving visualization (higher score == better binders).
- b. We revised our approach to compute the %rank (outlined in the MixTCRpred model's Methods section) to ensure consistency with the methodology used in other tools. This had no impact on the performance evaluation since the ranking of the predictions remained exactly the same.
- c. Following this, we re-did the plots of Fig.6 (predicting the TCRs-specific to SARS-CoV-2 epitope TFEYVSQLMDLE). The conclusions remain unaltered.

Reviewer #1 (Remarks to the Author):

The reviewers have addressed my previous concern by clarifying the TCR specificity prediction application scope. I think the conclusion will be helpful for readers to decide whether or not to use computational prediction methods. I would like to see this paper published.

We thank the reviewer for the positive comment. Following the comment of reviewer #4, we emphasized in the abstract the epitope-specific nature of MixTCRpred.

Reviewer #4 (Remarks to the Author):

I appreciate the authors' detailed responses to Reviewer 2's comments. The manuscript is clear with Nature Communications being an appropriate venue for publication. A few of Reviewer 2's comments center around the fact that MixTCRpred works the best for only specific epitopes rather than unseen epitopes, and I think some changes will further improve the manuscript. I base my comments on the numbering of the Reviewer 2's comments and the corresponding responses.

We thank the reviewer for the positive comment.

• [Major Comment 2 and 6] The authors should more prominently make clear in the abstract that MixTCRpred works with only specifically-trained epitopes. In the current version, it is mentioned in the last sentence, but given the authors' analyses on the pan-epitope network and the purposes of MixTCRpred, this should be more clearly defined.

We modified the abstract to emphasize the epitope-specific framework which is used by MixTCRpred.

- [Major Comment 8] In the same vein, if say practitioners were to have different epitopes than those trained, is it easy for them to train a version of MixTCRpred for their own use? If so, this is a great asset and maybe a tutorial on the software page will be beneficial.

The current version deposited on GitHub currently allows users to make predictions for 146 pMHCs using the MixTCRpred models trained on our manually curated dataset. We provide a google Colab tutorial to make predictions directly on the browser without downloading the repository. Moreover, we provide our curated dataset, which we think we'll be useful for the community. However, a user-friendly re-trainable version is currently not available, since retraining MixTCRpred is a less straightforward and more delicate process, requiring some expert knowledge in terms of choices of model architecture and hyper-parameters.

- [Major Comment 4] It may be better to mention this reasoning in the manuscript to discuss the choice of using the embedding layer as part of the network architecture.

We now briefly discuss the embedding as a part of the network architecture.

Reviewer #5 (Remarks to the Author):

The author introduced an important area of research, a vaccine design by machine learning. MHC interacting class I and class II allele diversity is absent as it is of utmost importance. Further epitopes variations should be introduced CD4 and CD8. While machine learning predicted with high accuracy there is a concern of overfit problem noted that reflects higher predictability. Importantly, TCR-pMHC recognition is highly dynamic, and machine learning alone cannot predict accurately; therefore, structural parameters should be included.